# A Potential of New Untreated Bio-Reinforcement from *Caesalpinia sappan* L. Wood Fiber for Polybutylene Succinate Composite Film

**DOI:** 10.3390/polym14030499

**Published:** 2022-01-26

**Authors:** Ekkachai Martwong, Yvette Tran, Nattawadee Natsrita, Chaithip Kaewpang, Kittisak Kongsuk, Yeampon Nakaramontri, Nathapong Sukhawipat

**Affiliations:** 1Division of Science, Faculty of Science and Technology, Rajamangala University of Technology Suvarnabhumi, Phra Nakhon Si Ayutthaya 13000, Thailand; ekkachai.m@rmutsb.ac.th; 2Soft Matter Sciences and Engineering, ESPCI Paris, PSL University, CNRS, F-75005 Paris, France; yvette.tran@espci.fr; 3Division of Polymer Engineering Technology, Department of Mechanical Engineering Technology, College of Industrial Technology, King Mongkut’s University of Technology North Bangkok, Bangkok 10800, Thailand; natthwd41@gmail.com (N.N.); chaithip4@gmail.com (C.K.); kittisak1487@gmail.com (K.K.); 4Sustainable Polymer & Innovative Composite Materials Research Group, Department of Chemistry, Faculty of Science, King Mongkut’s University of Technology Thonburi, Bangkok 10140, Thailand; y.nakaramontri@gmail.com

**Keywords:** biocomposites, polybutylene succinate, polymer films, fiber, cellulose, bio-reinforcement

## Abstract

Natural cellulose-based *Caesalpinia sappan* L. wood fiber (CSWF) has been demonstrated to have significant promise as a new untreated bio-reinforcement of the polybutylene succinate (PBS) composite film. The morphology, mechanical characteristics, and biodegradation were investigated. The morphology, the fiber distribution, and the fiber aggregation has been discussed. The properties of the composite have been improved by the addition of CSWF from 5 phr to 10 phr, while with the addition of 15 phr, the properties were dropped. The result showed that CSWF could be used as a new reinforcement without any treatment, and 10 phr of CSWF was the best formulation of a new biocomposite film. The PBS/CSWF10 composite film had the highest mechanical strength, with a tensile strength of 12.21 N/mm^2^ and an elongation at break of 21.01%, respectively. It was completely degraded by soil bury in three months. Therefore, the PBS/CSWF10 composite film has the potential to be a green with a promising short-term degradation.

## 1. Introduction

Petroleum-based packaging films, such as polyethylene and polypropylene, are a major source of pollution and have a negative impact on the environment [1,2]. Almost all plastic films are designed to be used once before being destroyed, resulting in a large volume of plastic film waste that poses a direct threat to the environment [3]. Thus, several researchers have attempted to develop a new solution to replace conventional plastic film with a fully biodegradable one.

Biodegradable plastic, also known as bioplastic, is defined by its ability to degrade naturally, such as polybutylene succinate (PBS) and poly(lactic acid) (PLA). PBS is mentioned as one of the possibilities appropriate for utilization as a bioplastic because of its outstanding mechanical characteristics, heat resistance, and processibility [4,5]. However, its properties are insufficient for daily use.

PBS has recently been enhanced by the use of inorganic materials [6]. However, it has a direct impact on humans, such as health problem explanation. Many research groups utilize cellulose in order to prevent these issues. It is a well-known natural-based additive used to improve the properties and biodegradability of several substances [7]. However, in order to improve the polymer matrix properties, cellulose-based fiber must first be treated before use, either chemically [8] or physically [9]. Almost every piece of research [10,11] has reported that treating cellulose with a chemical reagent or adding a compatibilizer enhanced the interfacial interaction between the polymer and natural-based cellulose. Finding a new bio-reinforcement without treating it or adding compatibilizers is a promising and desired alternative to save time and cost.

*Caesalpinia sappan* L. wood fiber (CSWF) is cellulose derived naturally from a Caesalpiniaceae plant. The mechanical characteristics are comparable to those of cellulosic materials. Brazilin and its derivatives, a heterotetracyclic structure with a hydroxyl functional group, are found in CSWF [12,13,14]. Thus, we hypothesized that CSWF may be a good reinforcement, even without any surface treatment or compatibilizers. Untreated CSWF is focus on being used as a bio-reinforcement for the PBS composite film to examine this hypothesis.

Therefore, the potential of untreated CSWF as a new reinforcement in a novel biocomposite film was investigated. The influences of CSWF content on the morphology, mechanical properties and biodegradability of PBS/CSWF composite films were studied.

## 2. Materials and Methods

### 2.1. Materials

PBS grade FZ71PM with a density of 1.24 g/cm^3^ and MFR (190 °C, 2.16 kg.) of 22 g/10 min was purchased from PTTMCC Biochem in Rayong, Thailand.

The CSWF originated from the OTOP center in Nan, Thailand. To obtain mesh size 100, CSWF was ground and sieved by mesh sieve size number 100. CSWF was less than 150 m long and 30 m in diameter. All of the components were dried for 12 h at 80 °C before use.

### 2.2. Method

#### 2.2.1. Preparation of PBS/CSWF Composite Films

To control the mixing conditions, a mixing temperature of 160 °C and a rotor speed of 50 rpm were utilized. The formula for PBS/CSWF composites is shown in Table 1. An additional minute was spent in an internal mixer preheating a PBS pellet. The CSWF was then added to the melted PBS at 0, 5, 10, and 15 phr and continuously mixed for 3 min. After that, the PBS/CSWF composite was dumped and rapidly cooled to ambient temperature. Finally, the composite was crushed by a granulator machine before being blown up.

The PBS/CSWF composite film was formed by blow film extrusion (type LF-400, Labtech, Sorisole (BG), Italy). Extruder temperature profiles from zone 1/zone 2/zone 3/zone 4 were 130/135/135/140 °C, respectively. The die temperature and screw rotational speed were 145 °C and 40 rpm, respectively. The nip-roll and windup rates were 4.2 and 4.4 mm/min, respectively. The lay flat width of composite films was fixed at 13 cm. To prevent humidity absorption, all of the composite films were kept in a desiccator.

An overview of PBS/CSWF composite preparation process is illustrated in Figure 1.

#### 2.2.2. Characterization

The density of the composite was determined using ASTM D4635-16. A Vernier calliper was used to accurately measure the length and thickness of composite films. The specimens were precisely weighed in order to calculate the density using the equation below.
Density (g/cm^3^) = Mass (g)/Volume (cm^3^)

The morphology of PBS/CSWF composite films was studied with an optical microscope at a magnification of 10× and 40×, using Xenon (DN-117M, Nanjing Jiangnan Novel Optics, Nanjing, China).

Following imaging, measurements of each sample were completed using ImageJ. The fiber size, aggregation area of fiber in PBS film composite and total area were analyzed. The statistical analysis of aggregation area, total area, and % area of fiber has been accomplished and reported.

The mechanical properties of the composite films were measured using Testometric (M500-25AT) (Testometric, Rochdale, UK) on a universal testing machine in accordance with ASTM D638. The size of the specimen was 130 × 10 mm^2^. The instrument included a 1 kN load cell and a 25 mm gauge length extensometer with a crosshead speed of 10 mm/min. Tensile strength and elongation at were reported.

Biodegradation was studied via soil burying in an ambient environment, and the weight change before and after soil burying was measured. The weight change was measured once a month for three months under ambient conditions. The Equation (1) was used to determine the film degradation.
W_loss_ (%) = (W_f_ − W_i_)/W_i_ × 100(1) where: W_loss_ (%) is weight loss percentage, W_i_ is the weight of the composite film before the burial test, and W_f_ is the weight of the composite film after the soil burial test.

## 3. Results and Discussion

### 3.1. Properties of PBS/CSWF Composite Films

The thicknesses of PBS/CSWF composite films were determined by setting the flat lay width to 13 cm as shown in Figure 2. It was discovered that the thicknesses of composite films for neat film, PBS/CSWF5, PBS/CSWF10, and PBS/CSWF15 were 0.031, 0.043, 0.052, and 0.069 mm, respectively. By increasing the CSWF component, the thickness of the PBS/CSWF composite film was marginally increased. It was attributed to agglomeration effects and limited dispersion in CSWF with a higher content.

Figure 3 illustrates the density of PBS/CSWF composite films. It was marginally reduced from 1.24, 1.20, 1.19, and 1.16 g/cm^3^ by increasing the CSWF content from 0, 5, 10, and 15 phr, respectively. Because CSWF has a lower density than neat PBS, the density tendency of PBS/CSWF composite film was therefore reduced in the formulation with a higher content of CSWF. As a result, by increasing the CSWF content, the density of the PBS/CSWF composite in this study was reduced significantly.

### 3.2. Morphology of PBS/CSWF Composite Films

Optical microscopy was used to examine the morphologies of PBS/CSWF composite films. Figure 4a shows OM images of PBS/CSWF composite films at 10× and 40× magnification. It was determined that CSWF was well-dispersed in the PBS matrix at 5 and 10 phr, with some CSWF aggregated at 15 phr. Furthermore, the results indicated that the composite film had no flaws between the PBS and CSWF interfacial surfaces. It was due to the polar–polar interaction and hydrogen bonding between PBS and Brazilin derivatives, an organic heterotetracyclic molecule found in CSWF. It was considered to be one of the reasons that enhanced the interfacial contact between PBS and CSWF. FTIR and TGA were two further techniques used to validate the interaction of PBS and fiber, as shown in Appendix A, respectively. Moreover, the ImageJ analysis findings were provided to access the behavior of the fiber aggregation.

Figure 4b presents the outcomes of fiber diameter and aggregation area, which are summarized in Table 2. The counts of fiber and aggregation point were 49, 58 and 51 for PBS/CSWF5, PBS/CSWF10 and PBS/CSWF15, respectively. By increasing the CSWF content from 5 to 15 phr, the overall aggregation area and average size of CSWF in PBS matrix rose from 3644 to 14,273 μm^2^ and 74.37 to 279.86 μm^2^, respectively. It was discovered that the area distribution was too high, and the SD value of all samples was greater than the average size of area. At this point, it was established that the distribution of fiber area was dramatically different, with PBS/CSWF15 showing the highest distribution of area. Furthermore, by increasing the fiber content from 5 to 15 phr, the cover area of the fiber in the film increases from 10.17% to 40.03%. The CSWF was entirely dispersed in the PBS matrix for the PBS/CSWF5 composite film, which included the least amount of CSWF and low area distribution. In contrast, the dispersion in the PBS matrix was low in the PBS/CSWF15 film, and the fiber was tightly aggregated in the PBS matrix confirmed by their statistical analysis of fiber area.

### 3.3. Mechanical Properties of PBS/CSWF Composite Films

The mechanical characteristics of the PBS/CSWF composite films were evaluated using tensile strength and elongation at break, as shown in Figure 5. They were studied in two directions: machine direction (MD) and transverse direction (TD). Figure 5a presents the tensile strength of composite films. The tensile strengths of composite films with 0, 5, 10 and 15 phr are 4.77, 5.69, 12.21 and 2.16 N/mm^2^ in MD and 2.21, 3.91, 6.51 and 0.43 N/mm^2^ in TD, respectively. Tensile strength tends to improve with increasing CSWF content. PBS/CSWF10 had the highest tensile strength in both directions, reaching 12.21 N/mm^2^ in MD and 6.51 N/mm^2^ in TD. In contrast, the mechanical properties of the PBS/CSWF15 composite film were significantly reduced in both directions.

Figure 5b presents the elongation at break results of the PBS/CSWF composite films. The elongations at break of the composite films with 0, 5, 10 and 15 phr are 10.21%, 11.11% 22.01% and 12.17% in MD and 3.68%, 5.07%, 5.53% and 4.19% in TD respectively. It resembled the tensile strength trend. PBS/CSWF10 had the highest elongation at break, with 22.01% of MD and 5.53% of TD, respectively. It could be described as a decrease in the PBS-to-fiber ratio and an increase in the aggregation area due to an increase in CSWF content from the OM image and ImageJ results. The optimal ratio and aggregation area of CSWF is 10 phr. As a result of CSWF overload and aggregation, the mechanical characteristics of PBS/CSWF15 were reduced. It is reasonable to conclude that an untreated CSWF has a high potential to be a bio-reinforcement.

### 3.4. Biodegradation Properties of PBS/CSWF Composite Films

The biodegradation process of the PBS/CSWF composite was monitored by soil burying for three months. The weight reduction of the composite films is shown in Figure 6a. The weights reductions of the composite films with 0, 5, 10 and 15 phr are 1.28%, 6.63%, 37.12% and 2.53% for the first month, 6.25%, 16.51%, 51.97% and 12.14% for 2 months and 19.33%, 42.85%, 100% and 13.1% for 3 months respectively. It was found that PBS/CSWF10 is the highest biodegradation rate. However, it was a surprise that the degradation rate of PBS/CSWF15 is lowest. This finding is supported and consistent with the mechanical properties. It might be because of the antibacterial abilities of Brazilin derivatives found in CSWF [12,13,14]. Composite films with CSWF contents ranging from 0 to 10 phr that released Brazilin derivatives onto the surface of the PBS/CSWF composite film were insufficient to inhibit the activity of bacteria attached to the composite film. Meanwhile, the PBS/CSWF15 film delivers sufficient Brazilin derivatives to the surface to inhibit bacterial activity. Figure 6b displays the appearance of composite films two months after degradation. According to the findings, PBS/CSWF10 degraded the fastest, whereas PBS/CSWF15 film degraded the slowest.

## 4. Conclusions

In the absence of treatment, CSWF was well-dispersed in the PBS matrix and could be employed as a new bio-reinforcement. The morphology, the dispersion, and the aggregation of fibers have all been validated and studied. The addition of CSWF increased the properties of the composite film from 5 to 10 phr, whereas the addition of 15 phr decreased the properties. After three months, the PBS/CSWF10 had completely degraded. PBS/CSWF10 is the best composite film, according to the results. Therefore, CSWF has the potential to be a promising alternative addition for the industrial sector of green plastic film.

## Figures and Tables

**Figure 1 polymers-14-00499-f001:**
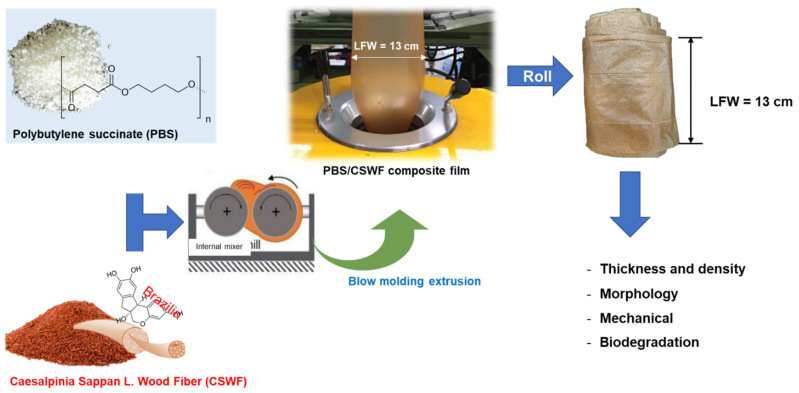
A short overview of PBS/CSWF composite preparation process.

**Figure 2 polymers-14-00499-f002:**
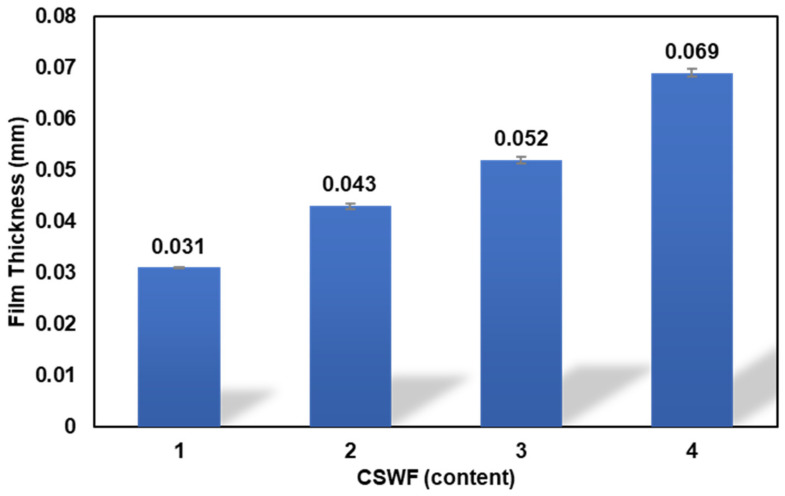
Thickness of PBS/CSWF composite films.

**Figure 3 polymers-14-00499-f003:**
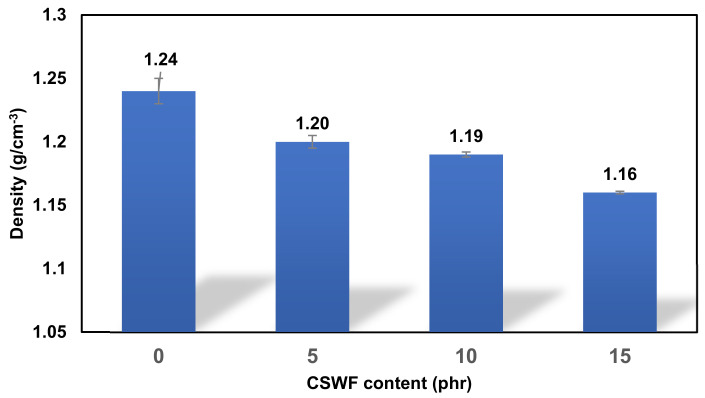
Density of PBS/CSWF composite films.

**Figure 4 polymers-14-00499-f004:**
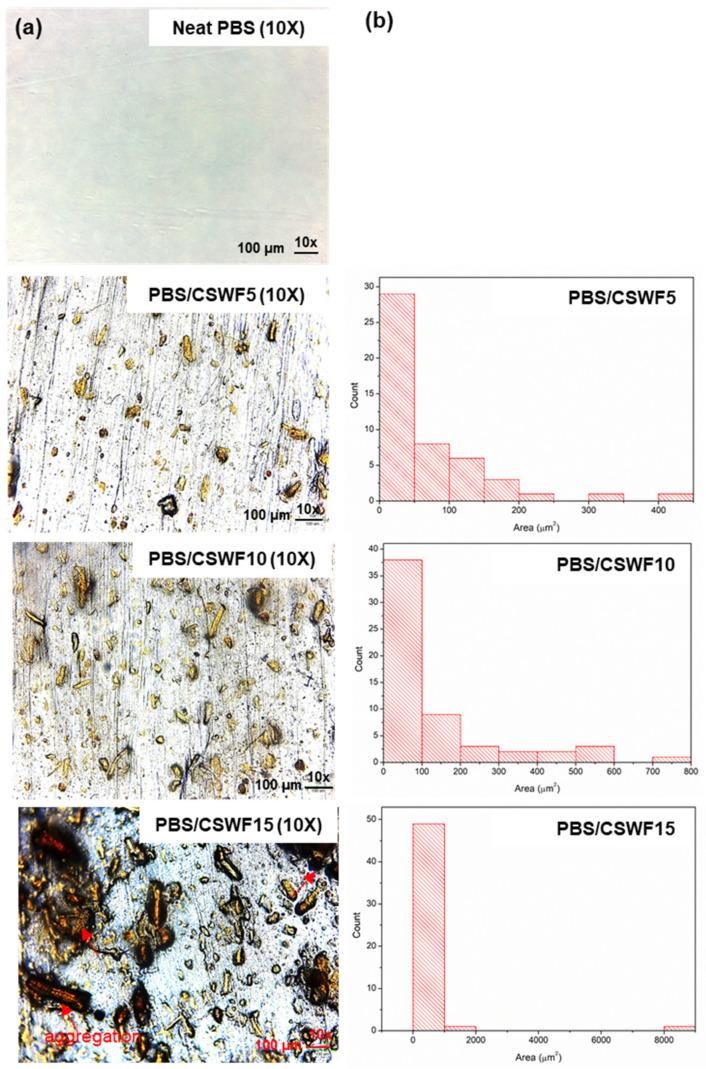
(**a**) Optical microscope images of PBS composite films at magnification of 10× and 40×, and (**b**) fiber diameter and its aggregation area analyzed by ImageJ technique.

**Figure 5 polymers-14-00499-f005:**
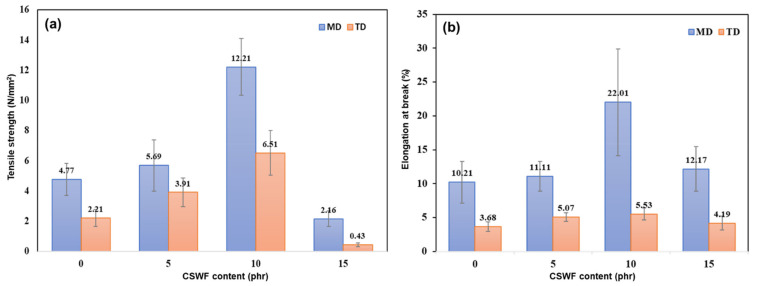
Mechanical properties of PBS/CSWF composite films—(**a**) tensile strength and (**b**) elongation at break.

**Figure 6 polymers-14-00499-f006:**
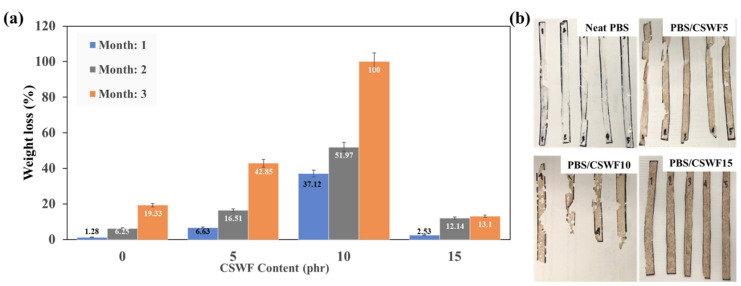
(**a**) The biodegradation progress of PBS/CSWF composite films with different burial times, and (**b**) The appearance of PBS/CSWF composite film after a 2-month biodegradation test.

**Table 1 polymers-14-00499-t001:** Compositions of the composite films.

Sample Name	PBS (phr)	CSWF (phr)
Neat film	100	0
PBS/CSWF5	100	5
PBS/CSWF10	100	10
PBS/CSWF15	100	15

**Table 2 polymers-14-00499-t002:** Summary results from ImageJ analysis.

Sample Name	Average Size (μm^2^)	Total Area (μm^2^)	Area (%)
PBS/CSWF5	74.37 ± 79.91	3644	10.17
PBS/CSWF10	126.26 ± 163.40	7323	20.22
PBS/CSWF15	279.86 ± 1145.97	14,273	40.03

## Data Availability

The study did not report any data.

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
