# Peer review of "A Potential of New Untreated Bio-Reinforcement from Caesalpinia sappan L. Wood Fiber for Polybutylene Succinate Composite Film"

_polymers, 2022, doi:10.3390/polym14030499_

Round 1
Reviewer 1 Report
Recommendation: Minor revisions needed.
Comments:
The paper by Martwong et al. contributes a Natural cellulose-based Caesalpinia Sappan L. wood fiber for the new untreated bio-reinforcement of the polybutylene succinate composite film. 10 phr of CSWF were found to be the best formulation of the new biocomposite film. The title and abstract are appropriate for the content of the text. The article gives an interesting scientific perspective on Natural cellulose-based Caesalpinia Sappan L. wood fiber material modification strategies.
Some issues should be addressed prior to publication.
- Page 6, Line 148 & Figure 4. “Furthermore, the results indicated that 148 the composite film had no flaws between the PBS and CSWF interfacial surfaces.” It would be helpful for the reader to understand how do you draw the conclusion and indicate the analysis of the optical microscope images. For example, what’s is the diameter of CSWF aggregated? How is the diameter change through the change of CSWF to PBS ratio? Also, please increase the font size of the scale bar on your images.
- Page 7, Line 165. “PBS/CSWF10 had the highest tensile strength in both directions, 164 reaching 12.21 N/mm2 in MD and 6.51 N/mm2 in TD. In contrast, the mechanical properties of the PBS/CSWF15 composite film were significantly reduced in both directions.” This data should be further discussed and explained. What is the rationale behind this behavior? Is this consistent with the optical image you showed in the previous section since you claim they are flawless? What happened when you increase the CSWF content from 10 to 15?
- Page 7, Line 173. “In this work, the mechanical properties of PBS/CSWF15 were reduced due to CSWF overload and aggregation. Therefore, the optimal content of CSWF is 10 phr.” The explanation is questionable. According to Figure 4, aggregation happened on both PBS/CSWF10 and PBS/CSWF15, that’s why I suggest you quantified the diameter change of the aggregation behavior via image analysis software such as ImageJ/FIJI. Otherwise, the conclusion is not supported by your optical image.
- Page 7, Line 181. “It was found that PBS/CSWF10 is the highest biodegradation rate. However, it was surprise that the degradation rate of PBS/CSWF15 is the lowest one.” This data is actually supported and consistent with your mechanical data. Also, please add “a” before the “surprise” for grammatical correction.
Author Response
On behalf of the authors, we would like to thank the editor and reviewers for their informative recommendations and remarks. We revised the manuscript in response to the suggestions. Our responses are included in the table below. In the revised manuscript, these revisions are tracked in change.

Reviewer 2 Report
Dear Authors,
This paper is devoted to studying the Composite films obtained by reinforcement of PBS with Caesalpinia Sappan L. fibers. Chemical characterization and biodegradation tests were performed. In order to improve paper quality, several issues must be addressed.
Title and keywords...the word "Fiber" must be included
Abstract
The methodology is not clear, does not focus only on the best treatment, describes results in a broad sense, and includes qualitative data also.
Introduction
It is suggested to include these fiber properties that make it interesting to be used as reinforcement.
Methodology
Mesh preparation is confusing... how do you reduce the particle size of fiber?
It is a lignocellulosic fiber? could you provide a physicochemical characterization?
Can you change the (phr) measure for another concentration unit in table 1.
Each methodology must be describe in detail, for instance, the composition of biodegradation test media.
Statistical analysis must be included in this section.
Results and Discussion section.
There is a lack of chemical characterization of both fibers (cellulose, lignin, hemicellulose) and composite (FTIR, TGA, SEM and/or XRD).
It is difficult and uncertainty to asses the morphological distribution of fibers in biocomposite using light microscopy only.
A proper discussion and comparison of your result with previous published work is need it on for each parameter.
Statistical analysis test must be included on each figure caption, when corresponding, and in discussion also.
Conclusion
Must be improved.
Author Response

(The authors gave the same response as above.)

Round 2
Reviewer 2 Report
Dear authors,
The statistical analysis is very important, and it must be included in the manuscript because it improves the discussion, not to remove it from it. Please, add it to your results.
Author Response
On behalf of the authors, we would like to thank the editor and reviewers for their informative recommendations and remarks. We revised the manuscript in response to the suggestions. Our responses are included in the table below. In the revised manuscript, these revisions are tracked in change.
Sincerely yours,
Nathapogs Sukhawipat
